# Vitamin D3 Inhibits the Viability of Breast Cancer Cells In Vitro and Ehrlich Ascites Carcinomas in Mice by Promoting Apoptosis and Cell Cycle Arrest and by Impeding Tumor Angiogenesis

**DOI:** 10.3390/cancers15194833

**Published:** 2023-10-02

**Authors:** Prashanth Kumar M. Veeresh, Chaithanya G. Basavaraju, Siva Dallavalasa, Preethi G. Anantharaju, Suma M. Natraj, Olga A. Sukocheva, SubbaRao V. Madhunapantula

**Affiliations:** 1Center of Excellence in Molecular Biology and Regenerative Medicine (CEMR) Laboratory, Department of Biochemistry, JSS Medical College, JSS Academy of Higher Education & Research, Mysuru 570015, Karnataka, India; prashanthkumarmv008@gmail.com (P.K.M.V.); chaithanyagowda55@gmail.com (C.G.B.); sivakumar65d@gmail.com (S.D.); preethiganantharaju@jssuni.edu.in (P.G.A.); suren156@gmail.com (S.M.N.); 2Department of Hepatology, Royal Adelaide Hospital, Port Rd, Adelaide 5000, Australia; olga.sukocheva@sa.gov.au; 3Special Interest Group in Cancer Biology and Cancer Stem Cells (SIG-CBCSC), Department of Biochemistry, JSS Medical College, JSS Academy of Higher Education & Research, Mysuru 570015, Karnataka, India

**Keywords:** vitamin D3, breast cancer, MCF-7 cells, MDA-MB-231 cells, Ehrlich ascites carcinoma, apoptosis, angiogenesis, cell cycle arrest

## Abstract

**Simple Summary:**

The burden of breast cancer has been increasing at an alarming rate. Epidemiological and empirical evidence has shown the key roles of vitamin D in the mitigation of tumor burden. But, the mechanisms by which vitamin D induces tumor reduction are poorly understood. Therefore, in the current study, we have aimed at understanding the role of vitamin D in inhibiting breast cancer progression in in vitro and in vivo models. The results of this study showed that vitamin D inhibited the proliferation of breast cancer cell lines in vitro and retarded Ehrlich ascites carcinomas in mice by inducing apoptosis as well as by halting cell cycle progression and angiogenesis. Future studies should focus more on strategies to improve the therapeutic index and the retention of vitamin D in tumor cells for better and long-lasting treatment.

**Abstract:**

The incidence of aggressive and resistant breast cancers is growing at alarming rates, indicating a necessity to develop better treatment strategies. Recent epidemiological and preclinical studies detected low serum levels of vitamin D in cancer patients, suggesting that vitamin D may be effective in mitigating the cancer burden. However, the molecular mechanisms of vitamin D3 (cholecalciferol, vit-D3)-induced cancer cell death are not fully elucidated. The vit-D3 efficacy of cell death activation was assessed using breast carcinoma cell lines in vitro and a widely used Ehrlich ascites carcinoma (EAC) breast cancer model in vivo in Swiss albino mice. Both estrogen receptor-positive (ER+, MCF-7) and -negative (ER-, MDA-MB-231, and MDA-MB-468) cell lines absorbed about 50% of vit-D3 in vitro over 48 h of incubation. The absorbed vit-D3 retarded the breast cancer cell proliferation in a dose-dependent manner with IC50 values ranging from 0.10 to 0.35 mM. Prolonged treatment (up to 72 h) did not enhance vit-D3 anti-proliferative efficacy. Vit-D3-induced cell growth arrest was mediated by the upregulation of p53 and the downregulation of cyclin-D1 and Bcl2 expression levels. Vit-D3 retarded cell migration and inhibited blood vessel growth in vitro as well as in a chorioallantoic membrane (CAM) assay. The intraperitoneal administration of vit-D3 inhibited solid tumor growth and reduced body weight gain, as assessed in mice using a liquid tumor model. In summary, vit-D3 cytotoxic effects in breast cancer cell lines in vitro and an EAC model in vivo were associated with growth inhibition, the induction of apoptosis, cell cycle arrest, and the impediment of angiogenic processes. The generated data warrant further studies on vit-D3 anti-cancer therapeutic applications.

## 1. Introduction

Despite extensive research and the implementation of various screening and preventive strategies against breast cancer (BC), incidence and mortality rates for this disease continue to grow [1,2]. BC is the most common cancer in women in developed and developing regions, with GLOBOCAN reporting approximately 2.261 million new BC cases and 0.684 million deaths in 2020 [3,4]. The insufficient accuracy of BC detection tests and the evolution of drug-resistant BC types, including triple and quadruple receptor-negative BCs, prompted the development of better diagnostic and treatment methods [5]. Multitargeting anti-cancer options (surgery, radiation, and estrogen deprivation therapy) are available for BC management, although these methods are often unsatisfactory in advanced stages of the disease [6]. Accordingly, the combination of anti-cancer drugs with natural supplements or compounds with low side effects and beneficial physiological characteristics is considered the most promising cancer prevention and treatment approach [7]. Several natural and essential nutritional compounds were shown to trigger the chemosensitization of endogenous mechanisms, including the activation of the immune defense network, and provide more efficient elimination of malignant cells [8]. Vitamin D represents a group of natural fat-soluble steroids with a described ability to inhibit the proliferation of malignant cells and trigger inflammatory reactions [9]. Vitamin D is highly effective in the form of cholecalciferol (or vitamin D3 (vit-D3), calcitriol). The biological actions of vit-D3 are mediated by vitamin D receptor (VDR), a ~48 kDa protein, a ligand-inducible transcription factor, which is expressed by nearly all cell types [10]. Vit-D3 binds to the VDR, thereby inducing dimerization with retinoid-X-receptor (RXR). The binding of the VDR-RXR complex to vit-D3 response elements (VDREs) in multiple gene-regulatory regions, located at promoters and the distal sites of target genes, triggers a range of biological responses [11]. Vit-D3 target genes were found to be involved in the control of cell proliferation, apoptosis, angiogenesis, and metastasis [12,13]. Additionally, several non-genomic pathways have been attributed to vit-D3-mediated anti-cancer activities. Advances in genomic and proteomic technologies have led to the identification of a variety of VDR target genes that mediate the antineoplastic actions of calcitriol [12,14]. Interestingly, VDR has been found to be elevated in many cancer cells [15,16], while a lack of vit-D3 was reported to be associated with the increased potential to develop cancers [17]. Vit-D3 insufficiency has been reported in the pathogenesis of various carcinomas, including gastric and breast cancers and non-Hodgkin’s lymphoma [18,19].

Recent studies have shown the importance of adequate nutrition in the management of cancers, especially BCs [20]. For instance, prior studies have reported that approximately one-third of cancer deaths are due to an imbalanced diet. Several studies have demonstrated that vit-D3 supplementation decreases the burden of BC in individuals lacking a sufficient concentration of the vitamin in serum (<20 ng/mL) [21]. As a potential anti-cancer agent, vit-D3 elevates the expression of p21 and p27, inducing cell cycle arrest [22] and triggering cancer cell apoptosis via the activation of caspase-3/7, p53, and Bax [23]. Several recent studies have also demonstrated the inhibition of cell migration by vit-D3. However, the role of VDR in the mediation of vit-D3 anti-cancer effects remains controversial and should be established in BC cells. Considering the high heterogeneity of BC cell types, it is unclear which BCs will be responsive to vit-D3-induced apoptosis. Multiple physiological and anti-inflammatory effects of vit-D3 support the therapeutical value of this agent. Therefore, it is essential to confirm and clarify the vit-D3-induced anticancer effects in vitro and in vivo.

In the present study, we tested VDR expression in a large set of BC cell lines. Notably, our data demonstrate elevated caspase-3/7, p53, and Bax proteins in BC cells treated with vit-D3, irrespective of VDR expression levels. Vit-D3 also arrested cell division in the G2/M phase, slowed the migration of cancer cells, and decreased angiogenesis. Furthermore, we investigated the molecular mechanisms of vit-D3 effects in vivo using mice models. Vit-D3 administration in vivo reduced the size of solid tumors and induced apoptosis in subcutaneously injected Ehrlich ascites carcinoma (EAC) cells. According to our detailed immunohistochemistry analysis, the observed vit-D3 anti-cancer effects were accompanied by very low systemic toxicity in vivo.

## 2. Materials and Methods

### 2.1. Materials

BC cell lines BT-474 (ER+, PR+, HER+, passages 40 to 50), MCF-7 (ER+, PR+, HER2-, passage #40–55), T47D (ER+, PR+, HER2-, passage #60–75), MDA-MB 453(ER-, PR-, HER2-, passage #70–90), MDA-MB-231(ER-, PR-, HER2-, passage #30–50), MDA-MB-468 (ER-, PR-, HER2-, passage #50–65), and lung cancer cell line A549 (passage #30–50) were obtained from the National Center for Cell Science, Pune, Maharashtra, India. Normal lung epithelial cell line BEAS-2B (passage #20–40) was provided by Dr. R. Thimmulappa (JSS Medical College, JSS Academy of Higher Education and Research, Mysuru, Karnataka, India). EAC cells were received from Dr. B.T. Prabhakar (Kuvempu University, Karnataka, India). Primary antibodies detecting VDR (cat#: VDR; D2K6W), p53 (cat#: 9282S), Bax (cat#: 5023S), and Actin (cat#:4970) were purchased from Cell Signaling Technologies (Danvers, MA, USA). Enolase (cat#: SC-7455), cyclin-D1 (cat#: SC-8396), and secondary antibodies (anti-Rabbit cat#: SC-2357 and anti-Goat cat#: SC-2020) were obtained from Santa Cruz Biotechnology (Santa Cruz, CA, USA). Antibodies for p53 (cat#: PM101), Ki67 (cat#: PM096), and CD31 (cat#: PR021) were from Path-n-Situ Biotechnologies Pvt Ltd. (Secunderabad, Telangana, India). FBS (cat#: 10270106) and PenStrep (cat#: 150763) were from Thermo Fisher Scientific (Waltham, MA, USA).

Vit-D3 (cat#: CP67970), diallyl disulfide (DADS) (cat#: SMB00378), cell culture grade DMSO (cat#: D2650), radioimmunoprecipitation assay (RIPA) buffer (cat#: R0278), protease inhibitor cocktail (cat#: S8820), bovine serum albumin (BSA) (cat#: 05479), camptothecin (cat#: 390238), itraconazole (cat#: I6657), and cisplatin (cat#: 1134357) were from Sigma Chemical Company (St. Louis, MO, USA). All cell culture plastics were from Techno Plastic Products (TPP) Pvt Ltd. (Bengaluru, Karnataka, India). Dulbecco’s modified Eagle medium (DMEM) with high glucose (4.5 g/L) (cat#: AL111), trypsin-EDTA (0.25%) (cat#: T001), Dulbecco’s phosphate-buffered saline (DPBS) (cat#: TL1OO6), ciprofloxacin (cat#: A032), acridine orange (AO, cat#: TC262), and ethidium bromide (EtBr, cat#: MB071) were from HiMedia Laboratories Pvt Ltd. (Bengaluru, Karnataka, India). Swiss albino mice were from Biogen Laboratory Animal Facility (Bengaluru, Karnataka, India).

### 2.2. Methods

#### 2.2.1. Vitamin D3 Cytotoxicity Assay

The cell growth-inhibiting activity of vit-D3 was measured using a previously described method [24]. BC cells (MCF-7, MDA-MB-231, and MDA-MB-468) were suspended in 100 µL DMEM supplemented with 10% FBS and seeded in a 96-well plate at a cell density of 1 × 10^4^ cells/well in 100 µL media. The cells were incubated at 37 °C with 5% CO_2_ and 90% relative humidity overnight. A stock of 100 mM of vit-D3 was prepared using 100% cell culture-grade (Hybri-Max sterile filtered cat#: D2650) DMSO. A 2× working stock of 1 mM was prepared by diluting the 100 mM Vit-D stock using DMEM with 10% FBS (The final concentration of DMSO was 0.5%). Further dilutions of vit-D3 were prepared by serial dilution using DMEM. The cells were exposed to increasing concentrations of vit-D3 at 7.78, 15.62, 31.25, 62.5, 125.0, 250, and 500.0 µM, for 24, 48, and 72 h. Cell viability was measured using a Sulforhodamine-B (SRB) assay [22]. The percentage of cell viability was calculated using the following equation: % Viability = 100 − (OD of Control − OD of Sample)/OD of Control) × 100. The experiments were repeated three times with at least four replicates for each concentration. 

#### 2.2.2. HPLC-Based Measurements of Vit-D3 Intracellular Uptake

Vit-D3 intracellular uptake was measured using HPLC [25]. MDA-MB-231 and MCF-7 cells (0.3 × 10^6^ cells/2.0 mL media/well) were plated in 6-well plates, left to grow for ~36 h, and treated with vit-D3 (25, 50,100, 200, 400 µM) for 48 h. Following this, the supernatant was collected, and proteins were precipitated using HPLC-grade methanol. The precipitated proteins were separated by centrifugation at 12,000 rpm. The collected supernatant samples (20 µL) were injected into HPLC. Vit-D3 was eluted (1.0 mL/min flow rate) from the C18 column (i.d. 4.60 × 250 mm) using HPLC grade methanol as a solvent system. The elution of compounds was monitored at 270 nm using a UV detector. A standard graph was prepared by plotting the area against the concentration of vit-D3. Extracellular vit-D3 concentration in the supernatant media was calculated. Cellular uptake was determined by subtracting the amount present in the supernatant from the total amount of vit-D3 used for treatment. 

#### 2.2.3. Measurement of DNA Integrity in Vit-D3-Treated Cells

MCF-7 and MDA-MB-231 cells (0.3 × 10^6^) were plated in 6-well plates and treated with vit-D3 (50, 100, 200 µM) for 48 h. After treatment, cells were trypsinized and centrifuged, and the collected cell pellets were resuspended in lysis buffer (Tris-HCl 10 mM (pH 8.0), EDTA 25 mM (pH 8.0), SDS 0.1%, Triton X-100 2%, NaCl 25 mM, distilled water). Following this, the cellular RNA was digested using RNase (incubation at 37 °C for 2 h). The cellular proteins were digested using Proteinase-K (100 µg/mL) at 50 °C for 90 min. The tubes were centrifuged, and the supernatants were combined with loading dye (6X-10 mM Tris-HCl pH7.6, 0.03% bromophenol blue, 60% glycerol, and 60 mM EDTA) for further analysis on 2% agarose gel. The samples were separated by running the gel at 50 V for 3 h. The separated DNA fragments were photographed using a gel documentation unit (SYNGENE G: G-Box Chemi-XR5, GENESys V1.4.1.0).

#### 2.2.4. Cell Death Assay with Acridine Orange and Ethidium Bromide (Et/Br) Staining

Cell death was assessed using cell staining with acridine orange (AO) and ethidium bromide (EtBr), as described previously [24]. MCF-7 and MDA-MB-231 cells (0.3 × 10^6^ cell/2.0 mL medium/well) were grown in 6-well plates for ~36 h and then treated with vit-D3 (50, 100, and 200 µM) or vehicle 0.5% DMSO (controls) for 48 h. Following this, cells were trypsinized to obtain a single-cell suspension. Trypsin was neutralized in a complete medium, and a 20 µL cell suspension was transferred to a different 1.5 mL microcentrifuge tube and incubated with EtBr (100 µg/mL) and acridine orange (100 µg/mL) mixture for 5.0 min. The stained cells were placed on slides and photographed using a fluorescence microscope (Olympus U-CMAD3) with fluorescein isothiocyanate (FITC) and tetramethylrhodamine isothiocyanate (TRITC) probes. Merged images with green (live) and orange (dead) cells were analyzed. 

#### 2.2.5. Cell Cycle Analysis

Cell cycle analysis was performed as described previously [26]. MCF-7 and MDA-MB-231 cells (2 × 10^5^ cells/2 mL/well) were cultured in a 6-well plate and treated with vit-D3 (50, 100, 200 µM) or vehicle DMSO control (1%) for 48 h. Media were removed and cells were washed with PBS and then trypsinized and centrifuged at 300× *g*, 25 °C for 5 min. The cells were then fixed in 1 mL of cold 70% ethanol for 30 min on ice, centrifuged, and the cell pellet was treated with 50 μL Ribonuclease-A (RNaseA; 100 µg/mL) for 4 h. Following this, the cell suspension was mixed with 400 μL PI solution (stock concentration 50 µg/mL)/10^6^ cells and incubated for 10 min at room temperature. Cell cycle distribution was measured using flow cytometry (BD FACS Calibur model: 343202-FACSCALIBUR 4 CLR, BD Biosciences, San Jose, CA, USA) with three filters (GFP-515/15, YFP-540/20BP, RFP—610/20BP). Cell quest pro v.6.0 software was used for cell analysis. 

#### 2.2.6. Caspase-3/7 Activity Assay

The caspase3/7 activity was measured using a fluorescence assay kit (cat# 10009135; Cayman Chemical Company, Ann Arbor, MI, USA). MCF-7 and MDA-MB-231 cells (3 × 10^5^ cells in 2.0 mL media/well in a 6-well plate) were treated with vit-D3 62.5, 125, 250 and 500 µM and vehicle (0.5% DMSO, controls) for 48 h. Cell lysates were collected using 200 μL of RIPA buffer (50 mM Tris HCl, 150 mM NaCl, 1.0% (*v*/*v*) NP-40, 0.5% (*w*/*v*) sodium deoxycholate, 1.0 mM EDTA, 0.1% (*w*/*v*) SDS and 0.01% (*w*/*v*) sodium azide at a pH of 7.4). Total protein content was estimated using a BCA kit as described previously [26]. Cell lysates (100 μg of total protein) were mixed with 100 µL of caspase-3/7 substrate N-Ac-DEVD-N-MC-R110 (200 μM final concentration). The reaction volume was increased up to 200 µL using 1X reaction buffer (20 mM HEPES, pH 7.4 containing 2 mM EDTA, 0.1% CHAPS, and 5 mM DTT). Caspase-3 was used as a positive control. The reaction mixture was incubated at 37 °C for 2 h and the caspase activity was measured using a Perkin Elmer multimode plate reader set at excitation = 485 nm and emission = 535 nm. The fold change was estimated by comparing vitamin D3-treated cells with vehicle-treated cell controls.

#### 2.2.7. Wound Healing (Cell Migration) and Angiogenesis Assays

MCF-7 and MDA-MB-231 (3 × 10^5^ cells in 2.0 mL media/well) were plated in 6-well plates and left to grow for ~36 h. When the culture dishes reached 95% confluence, a scratch was made using a 10 µL tip, as described previously [27]. Detached cells were removed carefully by washing with PBS. The cells were then treated with vit-D3 (100 µM) or vehicle 0.5% DMSO and photographed using an inverted microscope equipped with a Sony DSC-W710 camera. The area of the scratch was measured using Image-J software (NIH, Bethesda, MD, USA). Itraconazole (5 µM), a cell migration inhibitor, was used as a positive control [27].

The anti-angiogenic potential of vit-D3 (100, 200, 400 µM) was evaluated using a chicken chorioallantoic membrane (CAM) assay [28]. Fertilized eggs (day 1) were bought from Malavalli poultry (Karnataka, India). Eggs were wiped with 70% ethanol and incubated at 37 °C for the development of blood vessels. On day 7, a square (6.25 mm^2^) opening was punched using a single-edged razor blade. Eggs with viable embryos were selected and treated with 20 µL of vit-D3 (100, 200, 400 µM) or PBS (control). At least 3 eggs were used in each treatment group. The gap was closed using a parafilm followed by cello tape, and the eggs were placed back into the incubator for 24–48 h. The blood vessel size and number were assessed and photographed. 

#### 2.2.8. Western Blotting

To determine the impact of treatment with vit-D3 on the expression of VDR and key proteins involved in regulating cell growth and apoptosis, Western blotting was performed as detailed previously [29]. Whole cell lysates were collected from (ER+, PR+, HER2-) cell lines MCF-7 and T47D, (ER-/PR-/HER2+) cell lines MDA-MB-453 and SK-BR-3, and TNBC cell lines MDA-MB-231 and MDA-MB-468 [29]. MCF-7 and MDA-MB-231 (1 × 10^6^ cells/well) cells were treated with vit-D3 (62.5, 125, 250, and 500 µM) or with vehicle (0.5% DMSO) for 48 h. Cell lysates were collected using 200 µL of RIPA buffer. Total protein content was estimated using a BCA kit [30]. Cell lysate samples (50 µg of total protein) were loaded in 10% polyacrylamide gels with sodium dodecyl sulfate (SDS-PAGE) to be separated by electrophoresis as described previously [31]. Primary and secondary antibodies were diluted and used as recommended by the manufacturer. Protein bands were visualized using ECL and gel documentation system (UVI-TEC Alliance 9 System, UK). Protein expression (as band pixel density) was analyzed using Image-J software.

#### 2.2.9. Vit-D3 Effects in Swiss Albino Mice In Vivo

##### EAC Cell Growth in Mouse [Swiss Albino]

All animal experiments were approved by the Institutional Animal Ethics Committee, JSS College of Pharmacy, JSS Academy of Higher Education & Research, Mysuru, Karnataka, India (Approval #: No-155/PO/Re/S/99/CPCSEA; Karnataka, India). Female Swiss albino mice were bought from In vivo Biosciences (Bangalore, Karnataka, India). Animals were maintained according to CPCSEA guidelines. The mice (6–8 weeks old; 25–28 g) were divided into six groups as follows: Group I, control with no treatment; Group II, injected with EAC cells only (EAC/no treatment); Group III, injected with EAC and vehicle control 50% DMSO in PBS (EAC/vehicle) [32]; Group IV, injected with EAC and cisplatin (3.5 mg/kg in PBS; EAC/cisplatin) [33]; Group V, injected with EAC and 125 µg/kg vit-D3 (EAC/Vit D 125); Group VI, injected with EAC and 250 µg/kg vit-D3 (EAC/Vit D3 250) [34]. Viable (as determined by Trypan blue exclusion assay) EAC cells (1 × 10^6^/animal) were injected intraperitoneally. Six mice were used in each group. Cisplatin/vit-D3 intraperitoneal injections were repeated every other day for a total of 15 days. Body weight (in grams) was recorded every other day. On the 16th day, the mice were humanely sacrificed in a CO_2_ chamber and photographed. The ascites volume was measured, the total cell number was counted, and the number of dead cells was assessed. Peritoneal images were captured to observe anti-angiogenic effects. The vital organs were collected, fixed in formaldehyde, and processed for histochemical staining (hematoxylin and eosin (H&E) staining) to assess organ toxicity.

##### EAC Solid Tumors in Animal Model

A solid tumor mice model was adapted to determine the impact of vit-D3 on tumor growth in vivo as detailed by Kozokoshizuka et al., 1999 [35]. Female Swiss albino mice (6–8 weeks old; 25–28 g) were arranged randomly into 6 groups with 6 animals in each group. Following this, 2 × 10^6^ viable EAC cells were injected into the right thigh of 30 experimental animals (2 × 10^6^ cells/site/mouse; one site per mouse). Six animals in the control group were injected with the vehicle (50% DMSO in PBS). 

Tumor volume was measured using Vernier calipers once every five days after injection. From day 12, when the tumors reached a size of about 100 mm^3^, the mice were injected with cisplatin (3.5 mg/kg) or vit-D3 (125 µg/kg and 250 µg/kg) intraperitoneally. The treatment was continued every other day till day 24. On the 26th day, 3 mice from each group were sacrificed and tumors were collected. The remaining three mice were monitored continuously to determine the effect of the compounds on mean survival time and life span. The collected tumors were washed, weighed, and photographed. A portion of each tumor was fixed in formaldehyde and processed for H&E staining. Additionally, the sections were also stained for Ki67 and p53 markers using immunohistochemistry (IHC), as described previously [36]. Another portion of each tumor was used to collect total protein using RIPA buffer. Total protein content was estimated using a BCA kit, as described previously [30]. One hundred micrograms of the total protein was used to measure caspase-3/7 activity.

#### 2.2.10. Statistical Analysis

Cell viability experiments were conducted at least three times with at least three replicate measurements in each independent experiment. Results were expressed as the mean (of 3 independent experiments) +/− SEM calculated using Graph Pad Prism version 6.0. In vivo studies included 6 animals in each group. One-way or two-way ANOVAs were utilized to assess differences between vit-D3 and vehicle control groups. Tukey’s post hoc test was used after ANOVA and a “*p*” value of <0.05 was considered to be significant.

## 3. Results

### 3.1. Vitamin D Receptor (VDR) Expression and Vitamin D3 Effects in BC Cell Lines

In the majority of cases, vitamin D mediates its effects on cells by binding to VDR. Therefore, in order to assess the expression of VDR, an immunoblotting experiment was carried out by separating the whole cell lysates of breast cancer cell lines using SDS-PAGE followed by probing the transferred proteins with specific VDR antibody, as detailed in materials and methods. The analysis of immunoblotting data showed the relative VDR expression, which was found to be high in HER2-negative cell lines T47D, MDA-MB-231, and MDA-MB-468 (Figure 1). Low VDR expression was observed in HER2-positive cell lines SK-BR-3 and MDA-MB-453 (Figure 1). VDR expression was below the detection level in MCF-7 cells, although MCF-7 was defined as HER2-negative [37]. The data were reproducible in multiple experiments (Please refer to Appendix A for the Western blot and Appendix A for one of the replicates of the Western blot).

Prior studies have shown that vit-D3 induces cytotoxic effects in cancer cells [38]. To test HER2-positive and HER2-negative cells’ response to vit-D3 treatment, exponentially growing breast cancer cells were exposed to increasing concentrations of vit-D3 for 24 h, 48 h, and 72 h, and the viability of cells was determined by SRB assay. An analysis of the SRB assay data showed a dose-dependent decrease in the number of viable BC cells with the increasing concentration of vit-D3 in all tested cell lines (Figure 2). The calculated IC50 values are shown in Table 1. The growth of TNBC cell lines MDA-MB-231 and MDA-MB-468 was also inhibited by vit-D3 treatment with IC50 ranging from 150 µM to 250 µM over 24 h of exposure (Figure 2). Interestingly, even the triple-positive breast cancer cell line BT-474 was also inhibited by vit-D3 with an IC50 similar to other breast cancer cell lines (Please refer to Appendix A). Therefore, vit-D3 dose-dependently inhibited the BC cell growth independent of VDR status. This suggests that a low level of VDR expression may be sufficient to transmit vit-D3 effects in BC cells. 

To further test our hypothesis that very low VDR is sufficient to transport vit-D3, the cellular uptake of vit-D3 by BC cell lines MCF-7 (very low VDR) and MDA-MB-231 (moderate VDR) was determined using HPLC. Data analysis indicates 40% to 45% of vit-D3 uptake by both cell lines irrespective of VDR levels (Figure 3). These data suggest that either very low VDR is sufficient for vit-D3 absorption or the uptake of vit-D3 does not require VDR and may be associated with other mechanisms of intracellular transport. Additional studies are warranted to investigate the mechanism of vit-D3 cell transport in BC cells.

### 3.2. Vitamin D3 Inhibited MCF-7 and MDA-MB-231 BC Cell Cycle Progression

Vitamin D3 is known to arrest cancer cells in various phases of the cell cycle [23]. Even though many studies have reported G0/G1 blockade by vit-D3 treatment, few other studies have also reported the induction of G2/M arrest [22]. In our study, the MCF-7 cell line (low VDR) responded to vit-D3 treatment as shown in Figure 4A. Vit-D3-treated (200 µM) MCF-7 cells showed a significant increase in sub-G0/G1 cell population (an indicator of apoptosis, 46.66%) compared to vehicle-treated controls (2.57%). A simultaneous decrease in S- and G2/M cell populations was also observed (Figure 4A). The percentage of G0/G1 cell population was initially increased at the 50 µM dose of vit-D3 but decreased at the 100 µM and 200 µM doses (Figure 4A).

We have found that 200 µM of vit-D3 induced G2/M growth arrest in MDA-MB-231 cells (46.4%) compared to control cells (14.06%). A concomitant decrease in G0/G1 and S-phase cell populations was also observed in vit-D3 treated cells (Figure 4B). Used as a positive control, 25 µM camptothecin also induced a G2/M phase arrest while decreasing the percentage of cell population in G0/G1 and S-phases (Figure 4B). No significant changes in the sub-G0-G1 cell population were observed under these conditions (Figure 4B).

### 3.3. Vitamin D3-Induced Apoptosis Is Mediated by Caspase-3/7 Induction in MCF-7 and in MDA-MB-231 Cells

Caspase-3/7 activation is an indicator of the early stages of apoptosis [39]. A dose-dependent increase in caspase-3/7 activity was observed in MCF-7 cells treated with increasing concentrations of vit-D3 (from 62.5 µM to 500 µM) (Figure 5A). However, a noticeable decrease in caspase-3/7 activity was observed at 250 µM and 500 µM doses of vit-D3 in the MDA-MB-231 cell line (Figure 5A). This effect may be associated with advanced stages of cell death mechanisms represented by necrosis or other types of cell death at these high concentrations of vit-D3.

Increased DNA fragmentation indicates a late stage of apoptosis [22]. An analysis of DNA integrity with acridine orange and ethidium bromide [40] showed a significant increase in the late stages of apoptosis in cells treated with vit-D3. Vit-D3 (400 µM) treatment resulted in 52.0% of MCF-7 cell death compared to vehicle treatment (18%) (Figure 5B). A similar increase in cell death percentage was observed in the MDA-MB-231 cell line (Figure 5C). We observed fragmented DNA in cells exposed to 50 µM, 100 µM, and 200 µM of vit-D3 (Figure 6A). A similar DNA degradation pattern was observed even in MDA-MB-231 cells (Figure 6B).

### 3.4. Vitamin D3 Reduced BC Cell Migration In Vitro and Angiogenesis Ex Vivo

Increased cell migration reflects BC metastatic capacity [40]. An analysis of scratch assay data showed a significant decrease in the cell migration ability in the presence of 100 µM of vit-D3 in both MCF-7 (Figure 7A) and MDA-MB-231(Figure 7B) cell lines. Concentrations higher than 100 µM were also tested for anti-migratory effects but did not show in the photomicrographs as part of the anti-migratory effect, which could be because of increased cell death at these concentrations [41]. Tumor spreading is facilitated by angiogenesis [42], which is the formation of new blood vessels, to meet the nutrient requirements of fast-growing malignant cells. CAM assay data showed a significant decrease in the total number and size of the blood vessels upon treatment with Vit-D3 (Figure 8).

### 3.5. Analysis of Vitamin D3-Induced Effects on the Expression of p53, Cyclin-D, and Bcl-2

Elevated expression of tumor suppressor transcription factor p53 (an indicator of apoptosis) but decreased levels of cell cycle regulator cyclin-D1 (an indicator of proliferation) and Bcl2 (an indicator of cell survival) were detected in MDA-MB-231 cells treated with vit-D3 when compared to vehicle DMSO treated control cells; but, inconsistent changes in the expression of Cyclin-D1, Bcl2 and Bax were noticed in MCF-7 cells treated with vit-D3 (Figure 9A,B). An increased level of Bax was observed in vit-D3 treated MCF-7 cells but not in MDA-MB-231 cells (Figure 9A,B). Analysis of the Bax/Bcl2 ratio showed an increase in the ratio with 500µM vit-D3 treatment compared to control untreated and vehicle DMSO exposed cells in MCF-7; but no such increase was observed in MDA-MB-231 cell line (Figure 9A,B).

### 3.6. Vitamin D3 Effects on Tumor Growth In Vivo

The body mass of EAC-injected mice was increased as the time progressed from day 1 to day 15 (Figure 10A). Increased ascites volume and size of mice’s abdominal area were also observed with the increase in time (Figure 10B,C). However, vit-D3-treated mice demonstrated significantly decreased ascites volume (Figure 10C). A significant decrease in the formation of new blood vessels in the peritoneal membrane was also observed in mice treated with vit-D3 (Figure 10D). Moreover, vit-D3 enhanced the apoptotic cell population (Figure 10E). IHC examination showed no major differences in the architecture and overall morphology of liver tissues in vehicle- and vit-D3-treated mice (Figure 10F). No vascular degeneration or blood sinusoids were found in the vit-D3-administered mice (Figure 10G). However, a small focal area and capsular necrosis with chronic inflammatory cells were observed in the vit-D3-treated group. Thus, vit-D3 administration reduced the tumor burden in mice via the blockade of EAC cell growth in peritoneal fluid. These data suggest that vit-D3 may be used for the inhibition of the proliferation of circulating cancer cells.

Solid tumor growth was also decreased with vit-D3 administration, indicating that vit-D3 is an effective inhibitor of even solid tumors (Figure 11A–C). A significant increase in necrotic cell death was also observed in the tumor tissues of mice treated with vit-D3 (Figure 11D). The detected inhibition of the tumor growth was mediated by a significant decrease in Ki67-positive cells (a marker for cell proliferation; Figure 11E) and a non-significant increase in p53 expression (an indicator of apoptotic death; Figure 11F). In summary, vit-D3 is an effective anti-BC agent, which showed efficient apoptosis-inducing effects in vitro and in vivo. 

Vitamin D3 anti-cancer mechanism of action:



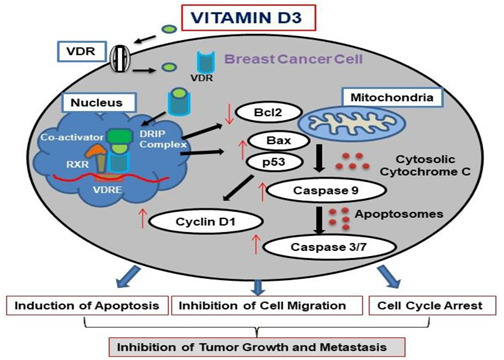



## 4. Discussion

Previous studies reported controversial information about VDR expression in different BC lines [41,43,44]. For instance, the MCF-7 cell line was both reported as VDR-negative [44] and -positive [43]. The MDA-MB-231 cell line was also reported as VDR-negative [44] and -positive [44] by different research groups. To clarify the role of VDR in mediating the pro-apoptotic effects of vit-D3 in BC, we tested the VDR expression in six different BC cell lines and identified VDR-positive and -negative BC lines. VDR expression was further correlated with ER, PR, and HER2 status in BC cell lines. We found that HER2-negative cells express higher levels of VDR compared to HER2+ cells. While our findings are in agreement with some of the previous data, the controversy remains. The existing divergence in VDR expression may be associated not only with the differences in cell line passages and maintenance but also due to variations in the assay used to measure the VDR expression (i.e., RT-PCR to measure the mRNA or Western blotting to measure the protein expression). We have observed that the majority of previous studies tested mRNA expression while neglecting VDR protein levels in BC cells. Notably, we detected only minor differences in the vit-D3 anti-proliferative effects between VDR-positive (MDA-MB-231) and VDR-negative (MCF-7) cells. The VDR-positive cell line had an IC50 of ~166 µM of vit-D3 (24 h of treatment), and VDR-negative cells had a similar IC50 of 168.6 µM of vit-D3. Longer treatment periods (48 h and 72 h) confirmed similar or reduced anti-proliferative vit-D3 effects in the tested BC cell lines. 

Supporting the observed effects, the preventive and therapeutic effects of vit-D3 were detected in various cancers, including breast carcinomas with variable levels of VDR [12]. Although it has been shown that vit-D3 effects in BCs depend on VDR, the receptor signaling network is highly heterogeneous, moderately unclear, and very cell-specific [12,45]. A recent study measured VDR expression in 718 invasive BCs using a tissue microarray and correlated the intranuclear and cytoplasmic receptor levels with decreased patient mortality [46]. Scored VDR expression was associated with both favorable patient outcomes and less aggressive tumor characteristics, including smaller tumor size, ER+/PR+, and a lower number of proliferating (Ki67+) cells. Other studies also indicated an important role of VDR in BC patients’ survival [46,47,48,49]. However, the mechanism of VDR-dependent vit-D3 effects remains to be clarified. For instance, vit-D3/VDR complexes were shown to impair and inhibit anti-cancer cytokine networks, which are responsible for the recognition and elimination of malignant cells, including tumor necrosis factor (TNF)/nuclear factor-ķB (NF-ķB) signaling systems [50]. However, the TNF network is also very complex and involved in the spreading of aggressive and resistant tumor cells, suggesting a dual role of vit-D3 in the regulation of pro-inflammatory pathways. Several rapid non-genomic vit-D3 effects were also observed in cells that do not express VDR [51]. Another study determined that vit-D3 diminished chemically induced carcinogenesis in a tissue-specific and VDR-independent manner in vivo [50]. Epigenetic changes in the VDR gene and/or the expression of VDR-regulated downstream genes should also be considered. Therefore, further studies are warranted to conclusively establish whether the BC cell line sensitivity to vit-D3 depends on a high level of VDR expression or if other mechanisms and/or membrane-localized VDR are involved. 

In our study, the treatment of BC cells with vit-D3 induced the expression of pro-apoptotic proteins p53 and Bax, G2/M cell growth arrest, reduced cell migration, and increased apoptosis (as demonstrated by elevated caspase-3/7 activity) in vitro. However, we have not tested the BC xenografts in mice, which may be considered a limitation of our study. Instead, EAC cells were used in our in vivo models. EAC cells represent mammary tumors and are widely used in cancer research [52]. Vit-D3 also retarded tumor growth in vivo via the inhibition of cell proliferation and the induction of EAC cell apoptosis. In the solid tumor model, vit-D3 reduced tumor size and tumor weight via the activation of necrotic cell death and the upregulation of caspase-3/7 and p53 expression. Supporting our findings, previous investigations demonstrated that vit-D3 treatment inhibited cancer growth via the elevated expression of cell cycle arrest proteins (p21 and p27) and increased activators and mediators of apoptosis, including caspase-3/7, p53, and Bax [53]. A separate study showed that cancer growth was retarded by vit-D3 via increased expression of the cell adhesion molecule E-cadherin and decreased transcriptional activity of catenin [54]. Vit-D3 was previously shown to reduce subcutaneous tumors (MCF-7 xenografts) in mice [55],and suppress growth-promoting pathways and genes, including MYC and estrogen receptors [56]. It has been shown that vit-D3-induced cell death was mediated by decreased Bcl2 levels but not due to increased p53 or caspase expression/activities [57]. However, in our study, we observed the activation of caspases and increased levels of p53 (Figure 9). The controversial data may be associated with cell-specific responses and differences in treatment times. We also observed that vit-D3 decreased cell migration and angiogenesis. The activation of angiogenesis is regulated by a group of angiogenic factors, including vascular endothelial growth factor (VEGF), interleukin-8 (IL-8), and macrophage inhibitory cytokine-1 (MIC-1) [58] However, in our study, we have not determined the expression of these molecules in the serum of control and experimental animals, which can be considered as a limitation of our study. Several previous studies have shown the vit-D3-dependent inhibition of cell migration via different effectors [59]. The role of the vit-D3/VDR axis in the regulation of tumor angiogenesis was linked to the expression of the BRCA1 gene [60]. Further studies are warranted to determine the impact of vit-D3 on the expression of pro-survival and pro-apoptotic markers in vivo.

In summary, we tested vit-D3 effects in a set of BC cell lines with different levels of VDR expression. Our data indicate that vit-D3 inhibited the growth of different BC cell subtypes triggering apoptosis and cell cycle arrest in vivo and in vitro despite differences in VDR expression. Vit-D3 also blocked angiogenic processes, thus facilitating the destruction of BC in vivo. Our findings support the beneficial role of vit-D3 as an anti-cancer agent in BCs. 

## 5. Conclusions

The results of this study demonstrated the anti-cancer potential of vit-D3 in six BC cell lines in vitro and in EAC-mouse models in vivo. The inhibition of BC growth and spreading was mediated by vit-D3-induced apoptosis, cell cycle arrest, and anti-angiogenic effects.

## Figures and Tables

**Figure 1 cancers-15-04833-f001:**
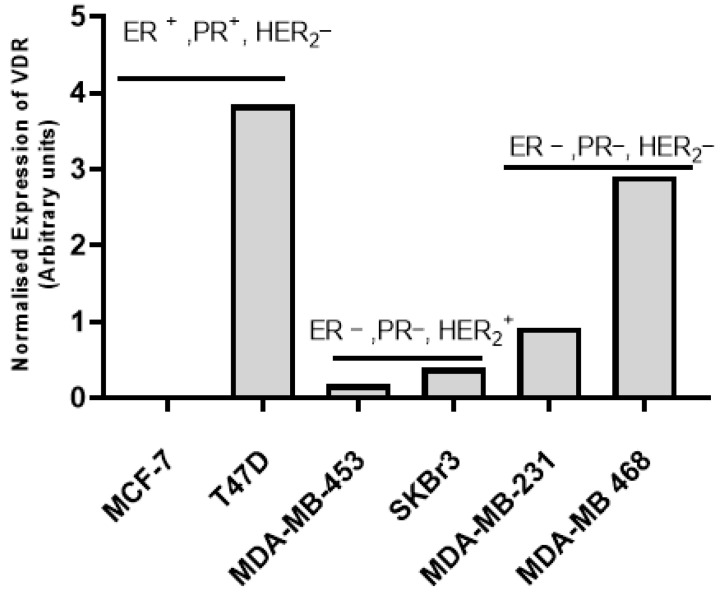
Immunoblotting analysis of VDR expression in BC cell lines. The expression of vitamin D receptor (VDR) was assessed in cell lines representing carcinomas of the breast by Western blotting. HER2-negative cell lines T47D, MDA-MB-231, and MDA-MB-468 expressed relatively more VDR compared to HER2-positive cell lines MDA-MB-453 and SKBR. Alpha-enolase served as a control for protein loading. The bar graph shows the normalized (intensity of VDR/intensity of enolase) expression of VDR. MCF-7, a HER2-negative cell line, had a very low VDR expression. The data represented are the average of 2 independent experiments.

**Figure 2 cancers-15-04833-f002:**
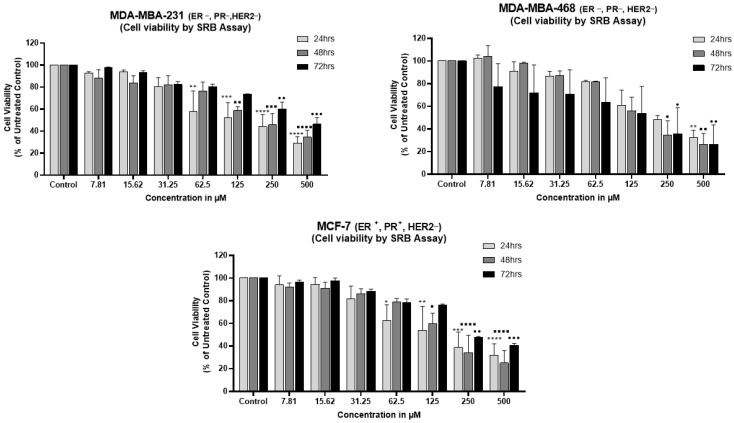
Vitamin D3 retarded the viability of breast cancer cell lines irrespective of VDR expression status. In order to determine the cytotoxicity profile of vitamin D3, first, breast cancer cell lines MDA-MB-231, MDA-MB-468, and MCF-7 were exposed to increasing concentrations of vitamin D and viability was determined by SRB assay at 24 h, 48 h, and 72 h of treatment. A dose-dependent decrease in the percentage of viable cells was observed in a dose-dependent pattern. But, the long-term incubation (47 h and 72 h) of cells with vitamin D had no enhanced effect (* *p* < 0.05; ** *p* < 0.01; *** *p* < 0.001; **** *p* < 0.0001; • *p* < 0.05; •• *p* < 0.01; ••• *p* < 0.001; ▪ *p* < 0.05; ▪▪ *p* < 0.01; ▪▪▪ *p* < 0.001; ▪▪▪▪ *p* < 0.0001).

**Figure 3 cancers-15-04833-f003:**
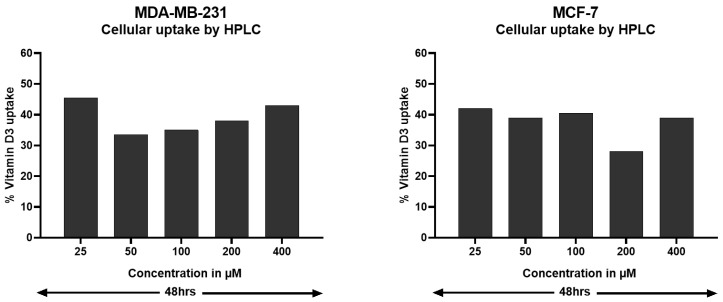
Analysis of the uptake of vitamin D3 by MCF-7 and MDA-MB-231 cells. In order to determine the amount of vitamin D3 taken up by the MCF-7 and MDA-MB-231 cells, a well-established HPLC method was used. After the treatment of BC cells, the vitamin D that remained in the culture medium was extracted using methanol, and its quantity was determined by injecting it into an HPLC C18 column. An analysis of the data showed about 40–50% uptake by BC cells. Increasing the concentration of vitamin D3 shows no significant increase in the uptake. The data represented are the average of 2 independent experiments.

**Figure 4 cancers-15-04833-f004:**
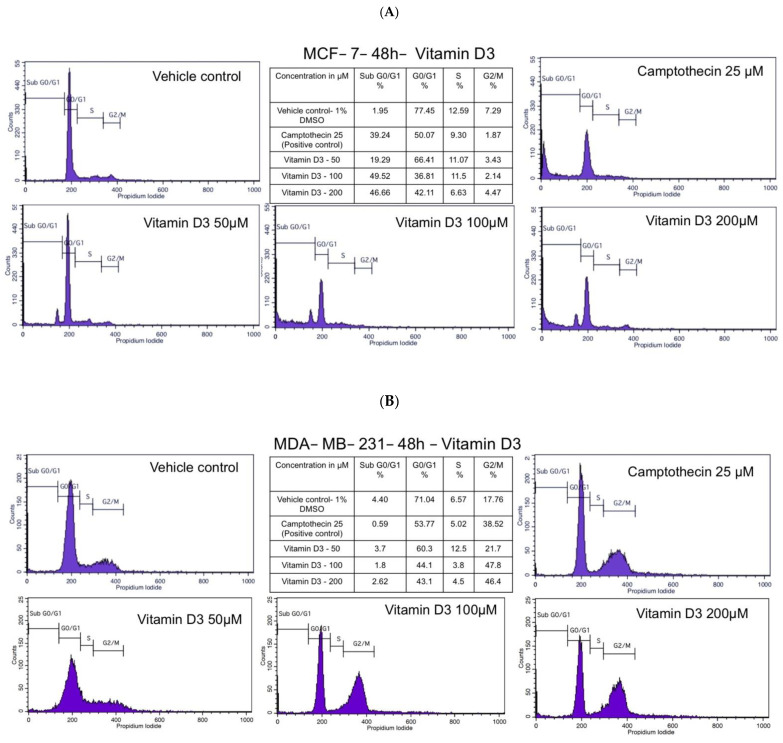
Cell cycle inhibition by vitamin D3. To check whether vitamin D treatment arrested cells in the G0/G1, S-, and G2/M phases of the cell cycle, the control and treated cells were stained with propidium iodide and analyzed by FACS. Vitamin D3 differentially modulated the cell cycle stages in a cell line-dependent fashion. (**A**) Low-dose vitamin D3 arrested MCF-7 cells in the G0/G1 phase and induced the accumulation of sub-G0-G1 cell populations. (**B**) Vitamin D3 arrested MDA-MB-231 cells in the G2/M phase and had minimal impact on sub-G0-G1 cells.

**Figure 5 cancers-15-04833-f005:**
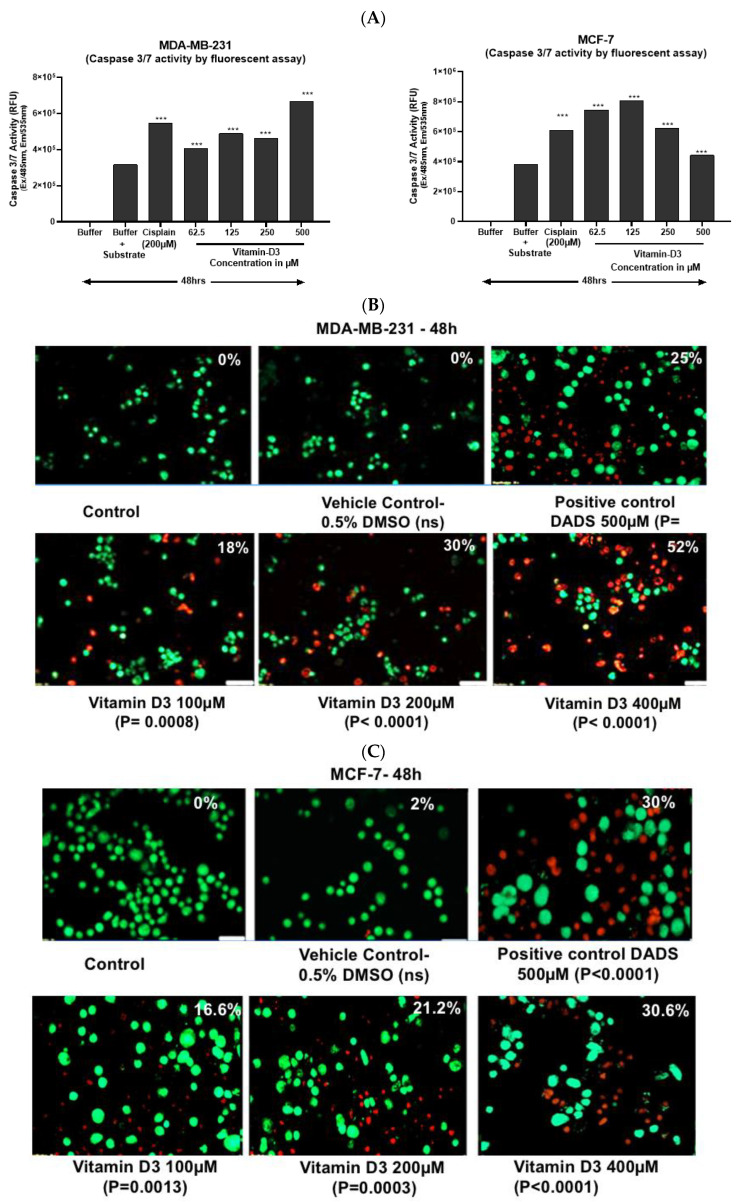
Vitamin D3 effects on caspase 3/7 in BC cells. Since caspases play an important role in the induction of apoptosis, the impact of vitamin D3 on the expression of caspase-3/7 was quantitated by using a fluorometric method, as detailed in Section 2. Vitamin D3 elevated the expression of caspase-3/7 in breast cancer cell lines in a dose-dependent fashion. Doses much higher than IC50 exhibited a decrease in caspase-3/7, which could be due to necrotic cell death at those concentrations. (**A**) Vitamin D3 treatment significantly increased caspase-3/7 activity in MDA-MB-231 and in MCF-7 cells, respectively. (**B**) Vitamin D3-induced apoptosis in MCF-7 cells. (**C**) Vitamin D3-induced apoptosis in MDA-MB-231 cells. The data represented are the average of 2 independent experiments and *p* values calculated using one-way ANOVA Tukey’s post-test. Significance is represented as *** with reference to control in caspase 3 activity (Mark the *p* value in Figure 6B; MDA-MB-231) (*** *p* < 0.001).

**Figure 6 cancers-15-04833-f006:**
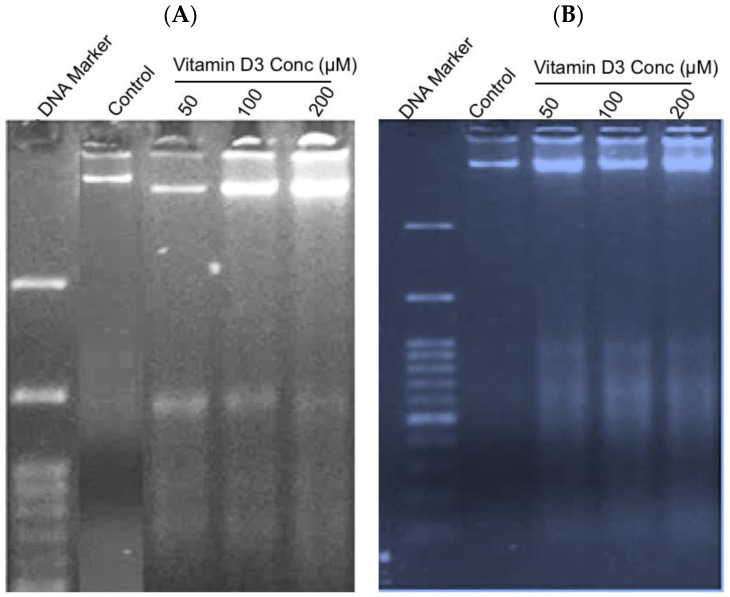
Vitamin D3-induced fragmentation of DNA: the fragmentation of DNA is another characteristic feature of cells undergoing apoptosis. Control and vitamin D3-treated cells were lysed and DNA was collected and analyzed using gel electrophoresis as detailed in Section 2.2. Vitamin D3-induced fragmentation of DNA in (**A**) MCF-7 cells and (**B**) MDA-MB-231 cells.

**Figure 7 cancers-15-04833-f007:**
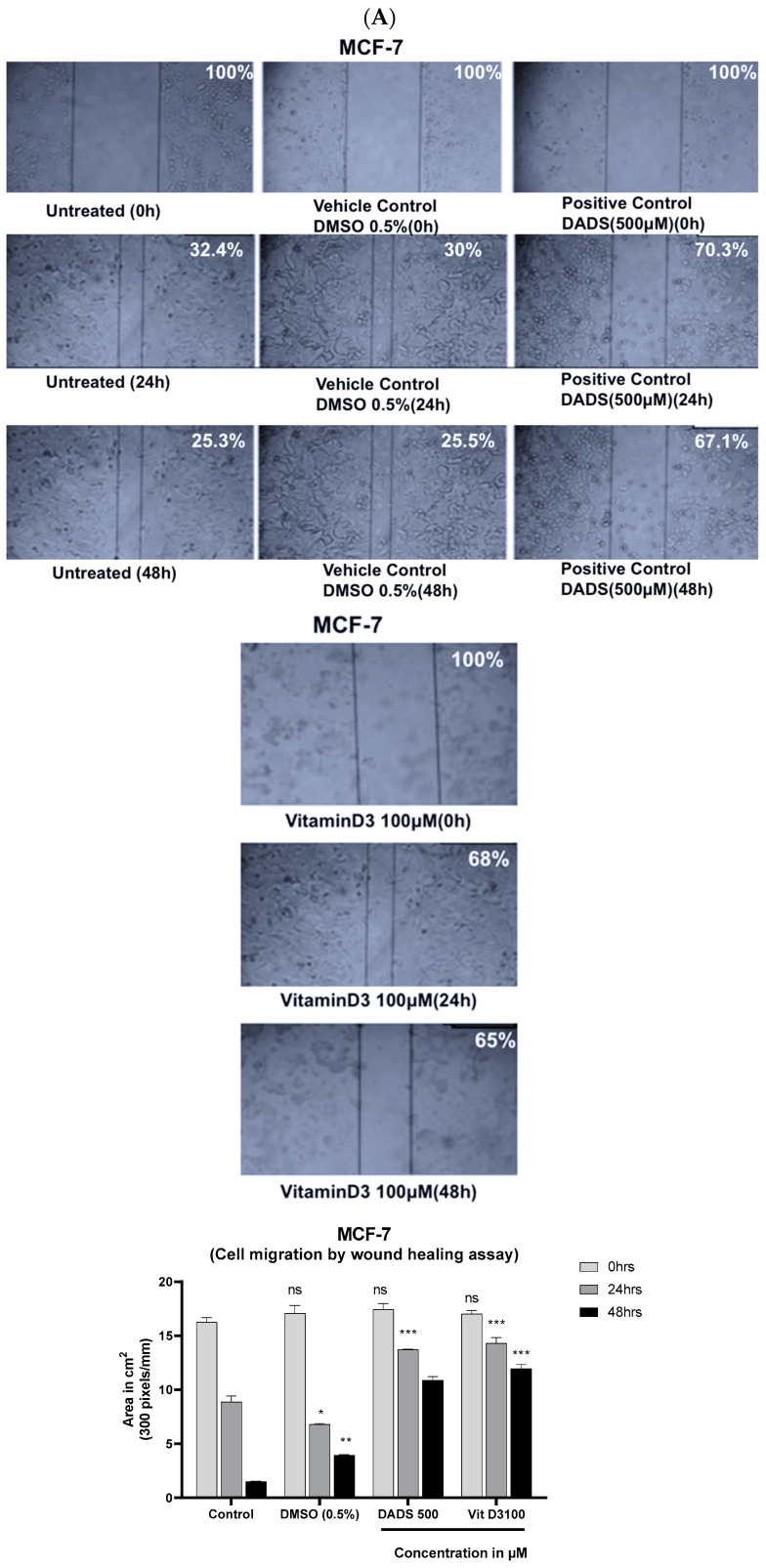
Vitamin D3 significantly decreased the cell migration ability at 100 µM concentrations at 24 and 48 h in MCF-7 (**A**) and MDA-MB-231(**B**) at 24 and 48 h. Statistical analysis using one way ANOVA showed that there were significant changes with Vitamin D and DADS treatment at 48 and 72 h (ns: Non-significant; * *p* < 0.05; ** *p* < 0.01; *** *p*< 0.001; ▪▪▪ *p* < 0.001).

**Figure 8 cancers-15-04833-f008:**
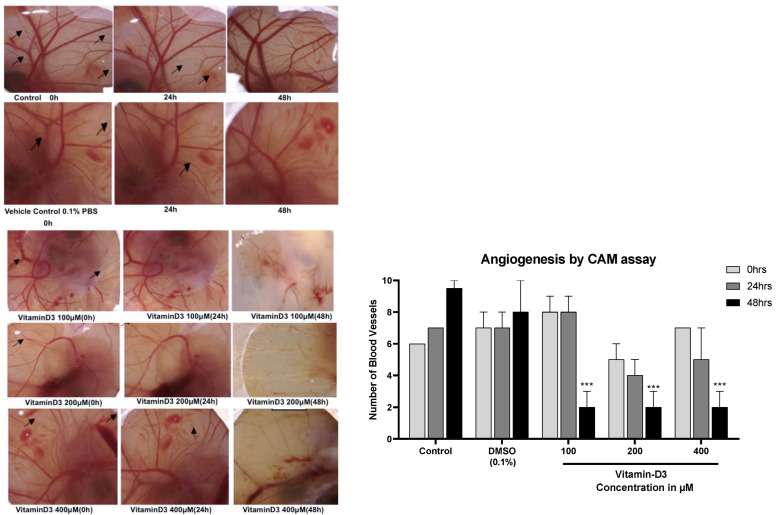
Vitamin D3 inhibited the formation of new blood vessels in the CAM assay: the CAM assay is one of the most appropriate models to study the angiogenic process. Treatment with vit-D3 inhibited the formation of new blood vessels, as evidenced by a significant decrease in the number as well as in the size of the blood vessel. Arrows in the image indicate the effect of drugs on blood vessel density. Statistical analysis using one way ANOVA showed that there were significant changes with Vitamin D treatment at 72 h (*** *p* < 0.001).

**Figure 9 cancers-15-04833-f009:**
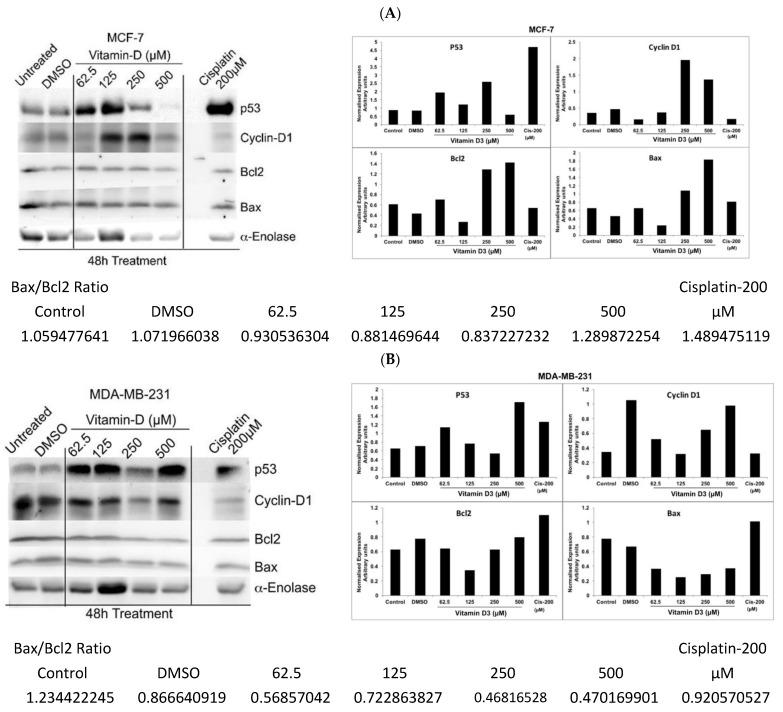
Vitamin D3 inhibited the proliferation of MCF-7 (**A**) and MDA-MB-231 (**B**) cells through the upregulation of p53 and (**B**) the downregulation of Bcl2. In order to determine the molecular mechanisms responsible for vitamin D3-induced cell growth inhibition, MCF-7 cells were treated with increasing concentrations of vitamin D3, and protein lysates were collected. Western blotting analysis showed a visible increase in p53 expression (an indicator of apoptosis) and a decrease in Bcl2 (an indicator of cell survival) upon the treatment of cells with vitamin D3. Alpha-enolase was used as a control for protein loading. Cisplatin (Cis-200) was used as positive control.

**Figure 10 cancers-15-04833-f010:**
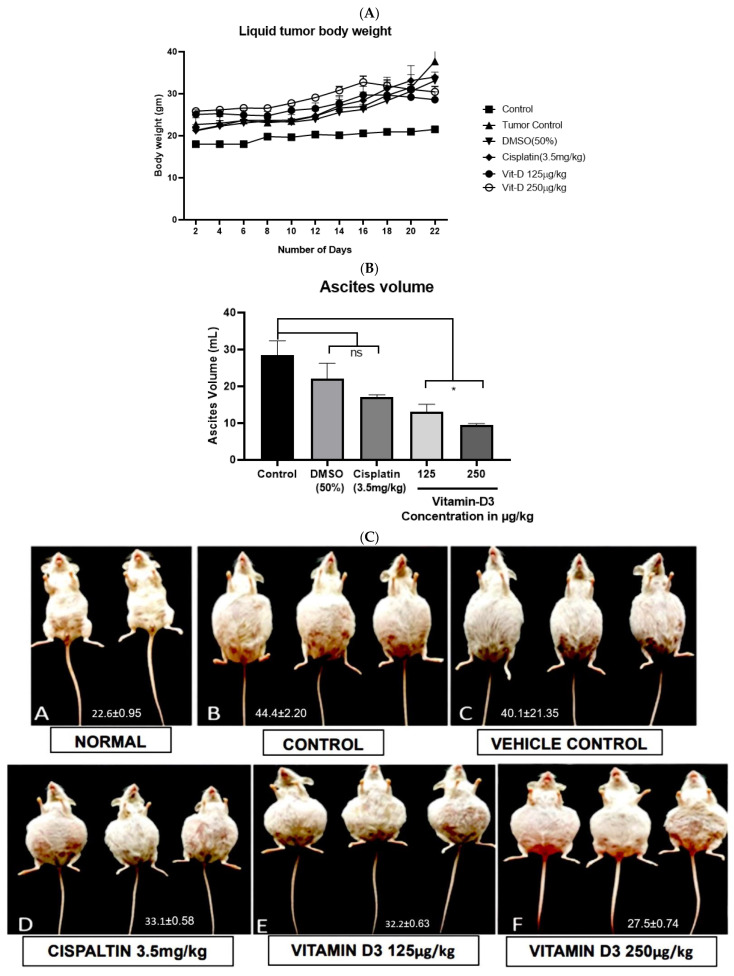
Vitamin D3 reduced the number of EAC cells in vivo. The intraperitoneal administration of vitamin D3 reduced the number of EAC cells present in the peritoneal cavity of Swiss albino mice. (**A**) Mice treated with vitamin D3 had much lower body weight (in grams), which is an indicator of liquid tumor growth; (**B**) vitamin D3 reduced the ascites volume in mice compared to those without treatment. (**C**) Photographs of normal and EAC-bearing mice indicate a visual difference in the size of the mice with vitamin D3 treatment: Top panel (**A**) normal; (**B**) control; (**C**) vehicle control; (**D**) positive control cisplatin; (**E**) vitamin D3 (125 µg/kg); and (**F**) vitamin D3 (250 µg/kg). (**D**) Vitamin D3 reduced the formation of new blood vessels in the intraperitoneal cavity of mice, indicating the anti-angiogenic effect. (**E**) The intraperitoneal administration of vitamin D3 induced apoptotic cell death in EAC cells. (**F**) The intraperitoneal administration of vitamin D3 is not toxic to the liver, as no visible differences were observed between the control and vitamin D3-treated mice. (**G**) Vitamin D3 induced capsular necrosis with chronic inflammation in the kidney. Arrows in the image indicate the effect of drug on blood vessel density. Statistical analysis using one way ANOVA showed that there were significant changes with Vitamin D treatment at 72 h (* *p* < 0.05; *** *p* < 0.001).

**Figure 11 cancers-15-04833-f011:**
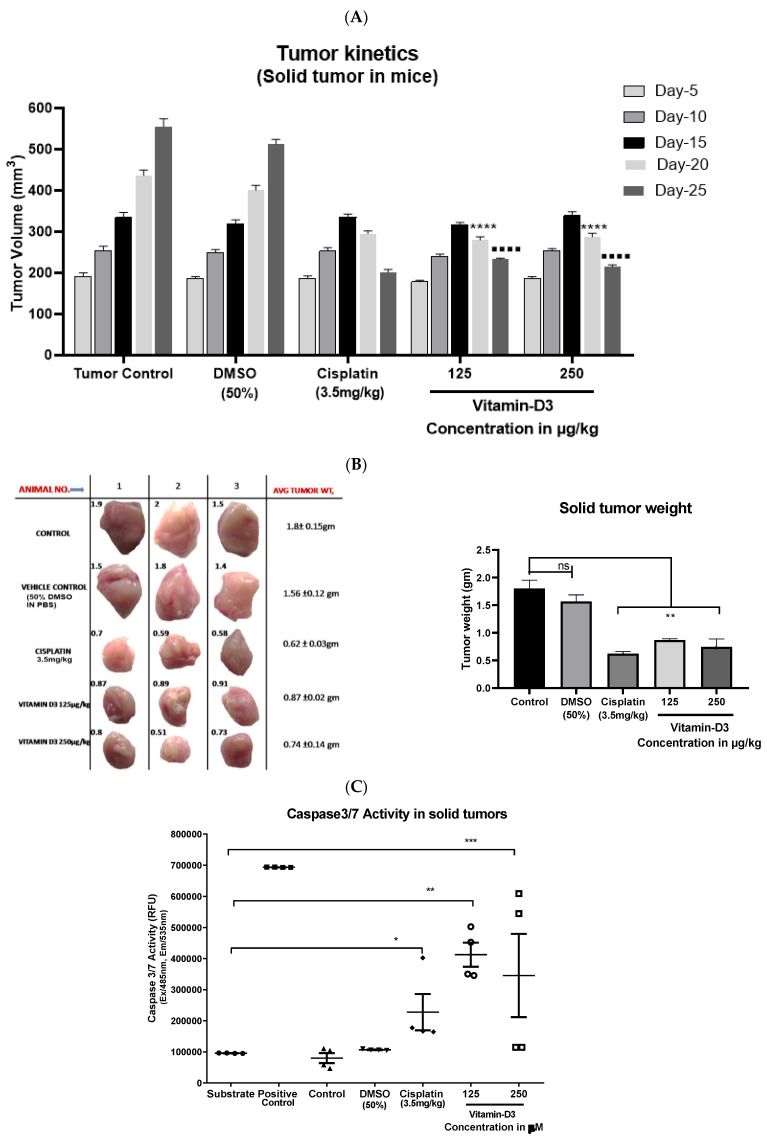
Vitamin D3 retarded the growth of solid tumors in Swiss albino mice. To determine the anti-tumor activity of vitamin D3, a solid tumor study was conducted as mentioned in Section 2.2. Vitamin D3 reduced the solid tumor volume (**A**); and tumor weight (**B**) in mice by increasing caspase-3/7 activity (**C**); and promoting necrosis (**D**). Furthermore, the intraperitoneal administration of vitamin D3 reduced the proliferation of solid tumor cells by reducing Ki67 expression (**E**) and by increasing the amount of p53-mediated apoptosis in solid tumors (**F**). Statistical Analysis was carried out using one-way ANOVA and the level of significance compared to control represented (ns: Non-significant; * *p* < 0.05; ** *p* < 0.01; *** *p*< 0.001; **** *p* < 0.0001; ▪▪▪▪ *p* < 0.0001).

**Table 1 cancers-15-04833-t001:** IC50 (µM) of vitamin D3 against BC cell lines at 24, 48, and 72 h.

Cell-Lines	24 h	48 h	72 h
MDA-MB-468	228.1 ± 52.41	169.7 ± 40.32	115.6 ± 35.54
MDA-MB-231	166.0 ± 59.04	230.2 ± 45.88	327.6 ± 22.521
MCF-7	168.6 ± 37.47	185.3 ± 38.04	297.1 ± 7.07

## Data Availability

The data that support the findings of this study are available from the corresponding author, upon reasonable request.

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
