# Peer review of "Vitamin D3 Inhibits the Viability of Breast Cancer Cells In Vitro and Ehrlich Ascites Carcinomas in Mice by Promoting Apoptosis and Cell Cycle Arrest and by Impeding Tumor Angiogenesis"

_cancers, 2023, doi:10.3390/cancers15194833_

Round 1

Reviewer 1 Report

1)    The concentration of DMSO used for the preparation of the increasing concentrations of vitD3 in “Vitamin-D3 cytotoxicity assay” should be mentioned in the M&M section.

2)    Why the authors did not use ethanol as a solvent for vitD3 since it was reported as better solvent for vitD3?

3)    Why the conc range of vit D3 used for the 2.2.2. setup is different than that used in 2.2.1. setup

4)    In 2.2.4, what was the vehicle and what was its concentration?

5)    Can the authors clarify why the used a cytotoxicity assay and cell death assay, both protocols may lead to the same conclusion?

6)    Why the authors used 0.1% DMSO as a control for the 2.2.6. and 2.2.7. setups, although they used 1% for the 2.2.5. setup?

7)    For the in vivo setup (2.2.9.1 – the numbering should be corrected as well), Why the authors decided to use these 2 concentrations specifically for vitD3? is there any reference? And, also, what are the references for the doses of other chemicals used in the same setup?

8)    For the solid tumors model (2.2.9.2 – the numbering should be corrected as well), I guess 50% DMSO as a control is too high! Most IACUCs suggest 0.5-5% DMSO, max at 10%. See link https://iacuc.wsu.edu/documents/2016/06/wsu_sop_1.pdf/

9)    Figure 1, authors should include the error bars for the bar charts.

10) Table 1, for all reported IC50, the authors should include the SEMs

11) Figure 3, authors should include the error bars for the bar charts. Also, could the author clarify why the 200 conc showed a decline in vitD3 uptake compared to other tested concentrations.

12) Figure 5A, the DNA fragmentation authors should include the error bars for the bar charts.

13) Figure 6, could the authors clarify why they did not investigate the effect of vitD3 on primary or near normal cells compared to MCF7 and MDA-MB-231 cancer cells.

14) For the CAM assay, I am not sure how vitD3 showed a significant decrease in total number and size of the blood vessels. This an ex vivo setup on a normal experimental model. In other words, generally the authors may point to vitD3 supplements may inhibit angiogenesis!!

15) Figure 9, the authors should calculate the Bax/Bcl-2 ratio. The ratio of Bax to Bcl-2 expression represents a cell death switch, which determines the life or death of cells in response to an apoptotic stimulus.

16) Figure 10A, I advise to have the limits of the y-axis between 15 to 40 as max for better representation of the line chart

17) The manuscript should be updated. Only one reference is listed from 2021 from the last 3 years!! (no refs from 2022 or 2023)

18) Generally, most of the figures are either small or of bad resolution.

Minor editing of English language required

Author Response

Query #1: The concentration of DMSO used for the preparation of the increasing concentrations of vitD3 in “Vitamin-D3 cytotoxicity assay” should be mentioned in the M&M section.

Response:  Details pertaining to the concentration of DMSO used for the preparation of the vitamin-D3 are provided in the revised submission.  A stock of 100mM vitamin-D3 was prepared in 100% DMSO, and diluted in a serial dilution manner to obtain a 2X secondary stock of vitamin-D3 in DMEM-Supplemented with 10% FBS. The concentration of DMSO in the 2X stock (beginning from 1mM) decreases by 2-fold as the vitamin-D3 concentration decreases.  The final concentration of DMSO in the treated wells is 0.5% as the well already has 100µL medium in the 500µM vitamin-D.  In the subsequent concentrations, the % DMSO decreases by 2-fold. 

Concentration of vitamin-D (in µM) in the well

500

250

125

62.5

31.25

15.62

7.81

Concentration of DMSO (%, v/v)

0.5

0.25

0.125

0.06125

0.0306

0.015

0.0075

Query #2:  Why the authors did not use ethanol as a solvent for vitD3 since it was reported as better solvent for vitD3?

Response: Authors do agree with reviewer for this comment.  The solubility of vitamin-D3 is much higher in ethanol compared to DMSO.  But, ethanol is much volatile compared to DMSO, hence there are chances of evaporation of the solvent, which might impact the final concentration of the drug.  More over, the LD50 of ethanol in mice, when administered intraperitoneally, is 1.65ml/Kg, which is compared to DMSO, whose LD50 is 6.2mL/Kg (Ref: Kelava, T., et al., Biological actions of drug solvents, Periodicum Biologorum, 113(3), 311-320, 2011).  Hence, DMSO is used as a vehicle to keep the effects of vehicles similar in experiments conducted in vitro and in vivo.  Several recent studies have also used DMSO as a vehicle for dissolving vitamin-D3 (Ref: Gharbaran, R., et al., BMC Res Notes 2019, 12, 216; Want, L et al., IJMS, 2020, 21(7), 2334).  DMSO is found to be non-toxic upto a max of 1% for majority of cancer cell lines (Ref: Gad, S.C. et al., Int. J. Toxicol., 2006.25(6), 499-521).  In our study, we have used 0.5% DMSO. 

Query #3:  Why the conc range of vit D3 used for the 2.2.2. setup is different than that used in 2.2.1. setup

Response: There is no specific reason for selecting these concentration ranges.  Since we have used the serial dilution method for all cytotoxicity studies to find the IC50, the concentrations have ranged from 500µM to 7.81µM (2.2.1).  In case of cellular uptake, since we know the IC50, we have selected the concentration range as shown in 2.2.1. 

Query #4:    In 2.2.4, what was the vehicle and what was its concentration?

Response:  The vehicle used was DMSO at 0.5% final concentration.  These details have been mentioned in the revised submission

Query #5:    Can the authors clarify why the used a cytotoxicity assay and cell death assay, both protocols may lead to the same conclusion?

Response:  Authors do agree with the reviewer that these assays measure the response of cells to cytotoxic agents.  Whereas the cytotoxicity assay, which was carried out by the addition of protein binding dye Sulforhodamine-B to cells, measures the total protein content, the acridine orange and ethidium bromide staining measures the DNA integrity.  Moreover, the AO/EtBr staining provides a visual representation and helps in classifying the cells in to live (green), apoptotic (orange) and necrotic (Red) cell types.

Query #6:    Why the authors used 0.1% DMSO as a control for the 2.2.6. and 2.2.7. setups, although they used 1% for the 2.2.5. setup?

Response:  Authors apologize for this confusion.  In all the assays the concentration of DMSO in 2X stock solution is 1% and the final concentration in the plate/well medium is 0.5%.  These details are incorporated in the revised submission

Query #7:    For the in vivo setup (2.2.9.1 – the numbering should be corrected as well), Why the authors decided to use these 2 concentrations specifically for vitD3? is there any reference? And, also, what are the references for the doses of other chemicals used in the same setup?

Response:  Two doses of vitamin-D3 were used to demonstrate and confirm that the anti-tumoral effects are due to vitamin-D3.  References for selecting the doses (125µg/kg and 250µg/kg) of vitamin-D3, and for other chemicals were added in the revised submission (page numbers 42)

Query #8:    For the solid tumors model (2.2.9.2 – the numbering should be corrected as well), I guess 50% DMSO as a control is too high! Most IACUCs suggest 0.5-5% DMSO, max at 10%. See link https://iacuc.wsu.edu/documents/2016/06/wsu_sop_1.pdf/

Response:  The number is corrected.  Fifty percentage DMSO has been used by many investigators previously (Please find one such reference in the revised submission, Page number 42). Even in our current study as well as previous reports from our laboratory, we have not observed any toxic symptoms with the 50% DMSO.  I hope the reviewer agree with this response

Query #9:   Figure 1, authors should include the error bars for the bar charts.

Response:  Figure 1, which shows the expression of VDR in breast cancer cell lines, by western blotting, the protein lysates were collected from two batches of cell lines and analyzed by western blotting.  The mean quantified data from two experiments is shown in the Figure 1.  Therefore, no error bar is added.  We have not observed much difference in these two replicate experiments, hence, did not repeat again

Query #10: Table 1, for all reported IC50, the authors should include the SEMs

Response: As suggested by the reviewer, the SEM for all the IC50 values was added

Query #11: Figure 3, authors should include the error bars for the bar charts. Also, could the author clarify why the 200 conc showed a decline in vitD3 uptake compared to other tested concentrations.

Response:  Figure 3, which shows the uptake of vitamin-D3 by breast cancer cells was also performed twice and the mean of two independent experiments is represented.  No much variation was observed between these two independent experiments.  At this point of time, we have no specific explanation why there is a decrease in the uptake of vitamin-D3 at 200µM concentration.  Additional studies measuring the uptake of vitamin-D at different time points at different concentrations might provide some clues.

Query #12: Figure 5A, the DNA fragmentation authors should include the error bars for the bar charts.

Response:  Figure 5A, is the data from Caspase-3/7 assay.  This fluorimetric assay was performed twice with at least 2 replicate wells per concentration.  Since the mean value is only from two independent experiments, no error bar is included.  We did not perform this experiment for the 3rd time as no much variation was observed between two independent experiments

Query #13: Figure 6, could the authors clarify why they did not investigate the effect of vitD3 on primary or near normal cells compared to MCF7 and MDA-MB-231 cancer cells.

Response:  The appropriate normal breast cell line MCF-10A is not available, hence, we could not test the effect of vitamin-D3 on this cell line. 

Query #14: For the CAM assay, I am not sure how vitD3 showed a significant decrease in total number and size of the blood vessels. This an ex vivo setup on a normal experimental model. In other words, generally the authors may point to vitD3 supplements may inhibit angiogenesis!!

Response:  Authors do agree with the reviewer that the CAM assay is a ex vivo setup in a normal experimental model.  Direct exposure of vitamin-D3 to angiogenic vessels of CAM showed decrease in the number and size of blood vessels.  Vitamin-D3 is known to inhibit tumor angiogenesis by targeting Nuclear factor kappa beta and interleukin – 8 signaling cascades.  Authors are not pointing that the vitamin-D3 supplements might inhibit angiogenesis.  Vitamin-D supplements are taken orally, hence are exposed to changes in the pH of fluids in gastrointestinal track.  Therefore, the effects due to direct exposure (as performed in this CAM assay) are likely to be different compared to the one that is exposed to several fluids with varied pH (when consumed through oral supplements)

Query #15: Figure 9, the authors should calculate the Bax/Bcl-2 ratio. The ratio of Bax to Bcl-2 expression represents a cell death switch, which determines the life or death of cells in response to an apoptotic stimulus.

Response:  As suggested by the reviewer, we have determined the Bax/Bcl2 ratio and incorporated in the revised submission (please refer Figure 9 in the revised submission).  We have observed an increase in Bax to Bcl2 ratio only in MCF-7.  But no such increase was observed in MDA-MB-231.  Potential reasons for such differences could be because of variations in the stability and turnover of these proteins in these two cell lines.

Query #16: Figure 10A, I advise to have the limits of the y-axis between 15 to 40 as max for better representation of the line chart

Response:  As suggested by the reviewer the limits of the y-axis were provided between 15 to 40 in the revised submission

Query #17: The manuscript should be updated. Only one reference is listed from 2021 from the last 3 years!! (no refs from 2022 or 2023)

Response:  As suggested by the reviewer, the references were updated in the revised submission.

Query #18: Generally, most of the figures are either small or of bad resolution.

Response:  The resolution and size of the figures are improved in the revised manuscript

Reviewer 2 Report

The manuscript by Veeresh et al. describes studies toward understanding the anti-cancer effects of vitamin-B3 and its potential as a preventative and/or treatment. The study focuses on breast cancer cell lines and uses representatives of variable hormone receptor status. The research includes both in vitro and in vivo experiments. The experiments described provide ample data on which the authors draw appropriate conclusions. The manuscript is well written with few typos (noted below). The discussion section provides context for the study related to the current state of the area of research.

This reviewer has a question with regard to the cell viability experiments. Table 1 indicates that the IC50 values for vitamin-D3 treatment increase (not expected) for longer treatment times of MDA-MB-231 and MCF-7 cells, but he IC50 value decreases (expected) for longer treatment times for MDA-MB-468 cells. Do the authors care to speculate on the origin of this phenomena?

This reviewer noticed a few typos:

Line 24; delete ‘as we as’ and add comma after ‘apoptosis’.

Line 350; period belongs on line 349.

Line 397; figure 6 caption need a period after ‘fragmentation’.

Figures 4, 9, 10 and 11 need better resolution.

Capitalization of journal titles in the references section is not consistent.

The English language usage in the manuscript is fine. Only a few typos were noted and addressed in the comments to the authors and editors.

Author Response

Query #1: This reviewer has a question with regard to the cell viability experiments. Table 1 indicates that the IC50 values for vitamin-D3 treatment increase (not expected) for longer treatment times of MDA-MB-231 and MCF-7 cells, but he IC50 value decreases (expected) for longer treatment times for MDA-MB-468 cells. Do the authors care to speculate on the origin of this phenomena?

Response:  Authors thank the reviewer for this comment.  The observed variations could be due to differences in the stability of vitamin-D3 as well as the expression of drug export proteins in these cell lines

Query #2: This reviewer noticed a few typos:

Line 24; delete ‘as we as’ and add comma after ‘apoptosis’.

Line 350; period belongs on line 349.

Line 397; figure 6 caption need a period after ‘fragmentation’.

Response:  The suggested changes have been incorporated in the manuscript

Query #3: Figures 4, 9, 10 and 11 need better resolution.

Response: The resolution of Figures 4, 9, 10 and 11 is improved in the revised manuscript

Query #4: Capitalization of journal titles in the references section is not consistent.

Response:   Authors have taken care of the capitalization of journal titles in the revised manuscript

Query #5: Comments on the Quality of English Language:  The English language usage in the manuscript is fine. Only a few typos were noted and addressed in the comments to the authors and editors.

Response:  As mentioned, we have addressed all the typographical errors in the revised manuscript

Reviewer 3 Report

Authors reported that the role of vitamin-D3 in breast cancer cells viability, apoptosis, cell cycle arrest and tumor angiogenesis in vitro. It has been widely studied that the functional role of vitamin-D3 in breast cancer progression and its deficiency is correlated with increased breast cancer risk. Finally, in vivo methods were utilized to address the anti-tumor efficacy of vitamin-D3 in mouse models and EAC breast cancer model. Administration of vitamin-D3 inhibited the growth of breast tumors in vivo suggesting that anti-tumor efficacy of vitamin-D3. In this study, authors mainly focused on the consequences such as cell viability, apoptosis, cell cycle arrest, angiogenesis and tumor growth rather than in depth molecular mechanism of anti-tumor efficacy of vitamin-D3. However, the important aspect of this study is authors utilized more than two breast cancer cell lines to show the anti-tumor effects of vitamin-D3. The following comments can further improve the manuscript.

Comments:

1. It would be important to quantify figures 5B and 5C and perform the statistical significance for figures 5A, 5B and 5C.

2. The quality of figures should be improved for figures 4, 5 ,6, 7 and 9.

3. Moderate grammatical changes and typos need to be corrected.

Moderate english language editing and typos needs to be corrected

Author Response

Comments:

Query #1: It would be important to quantify figures 5B and 5C and perform the statistical significance for figures 5A, 5B and 5C.

Response: Authors would like to thank the reviewer for this comment.  As suggested the quantification of figures 5B and 5C was performed and the percentage dead cells provided in the figure.  Statistical significance was performed and mentioned in the figure legend

Query #2: The quality of figures should be improved for figures 4, 5 ,6, 7 and 9.

Response: The quality of figures has been improved in the revised submission

Query #3: Moderate grammatical changes and typos need to be corrected.

Response: Typographical errors were corrected in the revised manuscript.  The manuscript is thoroughly read by the authors and necessary changes were made to avoid grammatical errors.

Comments on the Quality of English Language

Moderate English language editing and typos needs to be corrected

As mentioned, the manuscript is subjected to thorough reading to address all the issues related to grammatical errors

Round 2

Reviewer 3 Report

Authors addressed all the comments and improved the quality of the manuscript as suggested and therefore i recommend this manuscript for publication.